# Urea Ammonium Nitrate Solution Treated with Inhibitor Technology: Effects on Ammonia Emission Reduction, Wheat Yield, and Inorganic N in Soil

**Michael Thorstein Nikolajsen [1], Andreas Siegfried Pacholski [2] and Sven Gjedde Sommer [3,]***

[1]   University of Southern Denmark, Institute of Chemical Engingeering, Biotechnology and Environmental Technology, Campusvej 55, 5230 Odense, Denmark; M.michn14@student.sdu.dk
[2]   EuroChem Agro GmbH, Reichskanzler-Müller-Str. 23, 68165 Mannheim, Germany; andreas.pacholski@thuenen.de
[3]   Department of Engineering —Air Quality Engineering, Finlandsgade 12, building 98352, 8200 Aarhus N, Denmark
[*]   Correspondence: sgs@eng.au.dk

**Abstract:** Urea is the most used fertilizer nitrogen (N), and is often applied as urea ammonium nitrate (UAN), which may be an ammonia ($NH_3$) emission source after application. This study examined whether the addition of urease inhibitors reduced $NH_3$ emission, and, in combination with nitrification inhibitors, enhanced fertilizer N crop uptake. In three experiments, $NH_3$ emission was measured from plots (100 m$^2$) to which UAN was added with and without inhibitors. In March and May, the plots were covered with *Triticum aestivum* L., Sheriff (var), and in July, the soil was bare. The inhibitor mixed with urea was N-(n-butyl) thiophosphoric triamide (NBPT) and a mixture of NBPT and the new nitrification inhibitor DMPSA (3,4-Dimethylpyrazole succinic acid). Ammonia emissions were negligible from all plots after the first application of UAN due to the wet and cold weather while an average of 7% of applied UAN was emitted after application of UAN in April, where no significant effect of additives was observed. The harvest yield was low due to drought from May till August. Yield was highest when UAN was mixed with NBPT and lowest for untreated UAN. The highest emission from the bare plots was obtained from untreated UAN (17% of N), in contrast to 11% of N from the plots with added UAN + NBPT (not significant) and 7% from the plots with added UAN + NBPT + DMPSA (significantly different). Under the conditions of the current study, urease inhibitors reduce $NH_3$ emissions in periods where the risk of emission is high, and the combination of urease and nitrification inhibitors increased yields.

**Keywords:** UAN; urea; ammonia emission; crop yield; drought; urease inhibitor; nitrification inhibitor

## 1. Introduction

Urea (46% nitrogen content) is currently the most popular nitrogen (N) fertilizer, with about 80% of the world market for straight N fertilizers, and represents the major sectoral growth in the N industry (calculated from data in [1]). Urea is an uncharged molecule, which more readily infiltrates into the soil by diffusion or convection than positively charged ammonium ($NH_4^+$) [2]. After application, urea is hydrolyzed in soil to $NH_4^+$ within a few days [2]. The hydrolysis of urea increases soil pH, and this shifts the equilibrium between $NH_4^+$ to $NH_3$ towards $NH_3$, which causes emissions from applied urea to be up to 64% of applied N, and results in a low fertilizer efficiency [3]. The $NH_3$ emitted is a threat to human health, because it reacts with acidic compounds in the atmosphere, forming particulate matter, such as fine aerosol ($PM_{2.5}$), that causes lung diseases [4,5]. Ammonia-N deposited in land or waters may exceed the critical N loads of the ecosystems, causing eutrophication [6]. Several measures have

been developed to reduce the emissions from urea applied in the field, among which the slowdown of urea hydrolysis by the application of urease inhibitors (UIs) is one of the most robust [3]. Slowing down the hydrolysis by UIs allows time for urea to diffuse into wet soil or infiltrate into the soil after a rain event before hydrolysis occurs and this reduces the soil pH and enhances the absorption of $NH_4^+$, resulting in reduced $NH_3$ emissions.

In Denmark, urea made up only about 1% of the granular fertilizer used on Danish farms in 2014/15. In contrast, urea in liquid urea-ammonium-nitrate fertilizers (UAN) had a 17% share of the N fertilizer consumption in 2015 [7]. Ammonium fertilizers make up about 70% of the total fertilizer use in Denmark [7]. The modest use of plain urea is due to the risk of $NH_3$ volatilization and a resulting reduced reliability of the N fertilizer value. However, UAN also shows a high average potential of $NH_3$ emissions after field application (9.5%–12.6% of applied N [8]), which calls for mitigation measures, such as, e.g., the use of UIs.

A large number of studies have been carried out to develop and test UIs [3,9–14]. UI efficiency is related to climate and soil conditions, i.e., if rainfall follows within a few days after application of urea, the overall emissions from the untreated urea is low and there may be little or no quantitative effect of the inhibitors [15,16]. If only little rain falls immediately after urea or UAN application, the addition of inhibitors results in significantly reduced $NH_3$ emission [16,17].

Ammonium in soil is less exposed to N leaching than nitrate due to sorption to soil particles and to gaseous nitrous oxide ($N_2O$) and $N_2$ emissions following denitrification [14,18]. Nitrification inhibitors (NIs) slow down the transformation of $NH_4^+$ to nitrate ($NO_3^-$) after $NH_4^+$ fertilizer application or after urea hydrolysis to $NH_4^+$ in soil and this reduces the risk of $N_2O$ emission and $NO_3^-$ leaching. Still, the effect of the NI in fertilizer products may be variable as it is affected by the soil, crop, climate, or the mixing of additives, inhibiting hydrolysis of urea or nitrification [14,15,19–21]. Whether the addition of NI to urea and organic fertilizers can increase $NH_3$ emissions due to a prolonged time with high concentrations of $NH_4^+$ in the soil is also discussed [22]. To avoid negative effects of additives, the combination of adding NI with UI to urea is an option to control all loss pathways. The positive effects of this treatment of urea on both crop production and reduction of N losses by $NH_3$ emissions, $NO_3^-$, and $N_2O$ emissions has been shown in a wide array of studies [23]. However, only limited information exists on the efficacy of the combination of UI and NI in UAN. This may be due to the problem of potential low UI stability in UAN solution and a low global use of UAN compared to urea. In recent years, a new NI (3,4-Dimethylpyrazole succinic acid, DMPSA) was tested and has been shown to stabilize $NH_4^+$, and reduce $NO_3^-$ leaching and $N_2O$ emission losses from several base fertilizers (calcium ammonium nitrate, ammonium sulfate, urea) [24]. It is therefore of interest to test the reduction of $NH_3$ emissions from UAN with this new NI compound added in combination with the most common and robust UI (N-(n-butyl) thiophosphoric triamide, NBPT).

With the micrometeorological measuring method, unobtrusive measurements of $NH_3$ emission from field plots can be carried out, but in most studies, there are no replicates [15,18]. This is a severe shortcoming, as treatment effects cannot be statistically tested by replications. With wind tunnels, dynamic chamber studies, or the calibrated dynamic chamber method, $NH_3$ emission measurements are carried out with replicates [14,19–21,25], but this technology affects wind and rain, and is considered to only give a qualitative assessment of the effect of a treatment [14,19–21,25]. Furthermore, most studies of $NH_3$ emissions with the micrometeorogical method are from fertilizers applied to grassland or pasture [15,19–21], and the few measuring $NH_3$ emissions from cereals were carried out using wind tunnels or dynamic chambers [14,15,19–21,26]. To overcome the shortcomings of both methodological approaches, there is a need for a quantitative valid method of replicated $NH_3$ emission measurements.

Therefore, it is timely to examine the $NH_3$ emission from UAN in full-scale field plots with replicated treatments. A new multiplot micrometeorological method [27] was used to measure $NH_3$ emission from bare soil and from low and high winter wheat canopy, which are the times when UAN is applied to winter wheat. This new method will provide quantitative measurements of $NH_3$ emission from a replicated small plot, with the aim of catching the variation in $NH_3$ emissions from a field.

The objective of this study was to quantify $NH_3$ emissions from liquid UAN application without any additives, with UI (NBPT) added, and a combination of this UI and the NI (DMPSA) added. The fertilizers were applied to fallow fields and to winter wheat to quantify potential interactions of UI and NI on $NH_3$ emissions and assess the effect of no crop, and small and high canopy conditions on $NH_3$ emissions from UAN and UAN with inhibitors. In the study, the intention was to prove the hypothesis (1) that a multiplot micrometeorological method provides valid $NH_3$ emission measurement, and (2) test that treatment of UAN with UI reduces $NH_3$ emissions as efficiently as treatment of urea with UI, (3) test whether a combination of UAN with UI and NI reduces $NH_3$ emissions as the UI treatment in studies with urea, and (4) test if the reduction of $NH_3$ emissions and $NH_4^+$ stabilization with the two inhibitors provides a high crop yield and affects N species' ($NO_3^-$, $NH_4^+$) distribution in the soil.

## 2. Materials and Methods

$NH_3$ emission was measured during three periods from April till August 2018 within the growing season and after the harvest of winter wheat (*Triticum aestivum* L., Sheriff (var)) at Aarhus University's experimental research station in Aarslev, Denmark (55° 18 44.9 N, 10° 26 18.6 E). The soil is classified as a sandy loam (Typic Agrudalf), with 70% sand, 15% silt, and 13% clay, and a pH of 6.3 [28]. Aarslev is situated in a cool temperate climate, with an average annual temperature of 8.1 °C, annual total precipitation of 639 mm, and 1448 h of annual insolation (1961–1990 average; [29]). There were no protruding landscape elements south and south-west of the field, and to the east and north west was a 3 m high hedgerow (Figure 1). Wind and air temperature data measured at a 10-m height and rain from 1.5 m was obtained from the DMI climate station at Aarslev Experimental station situated 750 m from the field.

In August 2017, winter wheat 'Sheriff' (var.) seed was drilled in the field and not fertilized after spring barley (*Hordeum vulgare* L.), which was grown the foregoing two years on the field.

Ammonia emissions from plots with UAN added and unfertilized control plots were measured in March (before tillering (EC 19) on the BBCH crop development scale [30]), April (beginning of stem elongation, EC 29/30), and after harvest in August immediately after UAN solutions were applied to the plots. Soil samples from all the control and UAN-amended plots were collected before the application of UAN in March and after harvest in August before the application of UAN on the wheat stubble. Plants were harvested by hand in all control and UAN-amended plots in March and April, and with a plot harvester in August (Table 1).

**Table 1.** Overview of the plots and measurements included in the field trials.

| Date | Field and Plot | Measurements | Canopy |
|---|---|---|---|
| 30 September 2017 | Winter wheat 'Sheriff' (var.) seed drilled in the field | | |
| 13 March 2018 | | Soil sampling: 0–30 cm, 30–60 cm and 60–90 cm all 16 plots. | |
| **Experiment 1** | | | |
| 23 March 2018 | 43 kg N ha$^{-1}$ applied to plots in UAN except for the control plots not added N. Treatments; UAN, UAN + UI*, UAN + UI* + NI** P and K applied at rates corresponding to the amounts in the field around the plots, i.e., 20 kg P ha$^{-1}$ and 60 kg K ha$^{-1}$ | $NH_3$ emission measured from 23 March till 4 April | Wheat plants 5 cm, physiological growth stage 10–12 |
| 14 April | 120 kg N ha$^{-1}$ applied to the field (Except plots) in liquid mineral fertilizer NPK 15-2-6 with the N being in the form of urea (Amid) and included 2% sulphur (S). | | |
| 17 April | | Plant harvest in all plots and plant physiology age characterized | Wheat plants 5–10 cm physiological growth stage 15 |

**Table 1.** *Cont.*

| Date | Field and Plot | Measurements | Canopy |
|---|---|---|---|
| | **Experiment 2** | | |
| 20 April–5 May 2018 | 140 kg N ha$^{-1}$ applied to plots in UAN except for the untreated control plots not added N. Treatments; UAN, UAN + UI*, UAN + UI* + NI** | NH$_3$ emission measured from 20 April till 5 May 2018 | |
| 16 May | 80 kg N ha$^{-1}$ applied to the field (Except plots) in mineral fertilizer (NPK 15-2-6 with the N being in the form of urea (Amid) and included 2% S. | Entire field except plots | |
| 22 May 2018 | | Plant harvest in all plots and plant physiology age characterized | Wheat plants physiological growth stage 37–41 |
| 25 July 2018 | | Harvest with plot combine harvester all plots. Measured the yield and N uptake (grain and straw) | |
| 16 August 2018 | | Soil sampling in plots: 0–30 cm, 30–60 cm and 60–90 cm all 16 plots. | |
| | **Experiment 3** | | |
| 16 August 2018 | 100 kg N ha$^{-1}$ applied to plots in UAN except for the untreated control plots not added N. Treatments; UAN, UAN + UI*, UAN + UI* + NI** | NH$_3$ emission measured from 16 August till 27 August 2018 | Harvested field, stubble field. |

*UI; urease inhibitor (N-(n-butyl) thiophosphoric triamide, NBPT), dosage 0,47 g/l UAN. **NI; nitrification inhibitor (3,4-Dimethyl pyrazole succinic acid, DMPSA) dosage: 2.18 g/l UAN.s

## 2.1. Experimental Layout and Fertilization

All NH$_3$ measurements in replicated plots were arranged in a randomized, replicated complete block design, with three or four replications of treatments on square plots with a side length of 10 m (P10 plots). As a quantitative control of the small plot measurements, one standard plot was used for micrometeorological measurements with a side length of 36 m (P36) to which UAN without inhibitors was added. The plots were separated by a minimum of 40 m to avoid cross-contamination (Figure 1).

The three treatments investigated on the P10 plots in each of the three experiments were the control with no fertilizer applied, UAN, UAN + UI (NBPT), and UAN + UI (NBPT) + NI (DMPSA) (Table 1). In the first and second application of fertilizers in March and April, 16 square plots with a side length of 10 m (P10) were laid out in the field and there were four replicates of each treatment. In the August experiment, nine square plots with a side length of 10 m were used in the tests and there were three replicates of each treatment. To the P10 plots, 43 kg N ha$^{-1}$ in liquid UAN was added on 23 March, 140 kg N ha$^{-1}$ was added on 20 April 2018, and 100 kg N ha$^{-1}$ on 13 August 2018.

In March and April, UAN was applied in a square plot with a side length of 36 m (P36) simultaneously with UAN application in the P10 plots to compare the small plot-based approach with measurements of the standard mass balance approach employing passive flux samplers.

In the March trial (Exp. 1), the liquid fertilizer was applied manually to the P10 plots using a pesticide sprayer, and the plots were subdivided in squares to facilitate a homogeneous application rate. To the P36 plot, in March, the liquid solution was applied with a tractor-driven pesticide spreader, with each nozzle emitting three bands of UAN per nozzle. In the two following experiments (Exp. 2 and 3), the liquid fertilizer was applied to all plots with the tractor-driven insecticide sprayer (Table 1).

To avoid island effects by an unfertilized canopy surrounding the P10 plots, the field surrounding the experimental plots was fertilized with 120 kg N of liquid mineral urea fertilizer, NPK 15-2-6, on April 14 2018, and 80 kg N of NPK 15-2-6 was added on 16 May. Winter wheat was harvested by a plot combine harvester on 25 July 2018. All experimental plots were supplied with P and K fertilizers to fulfil the crop demands.

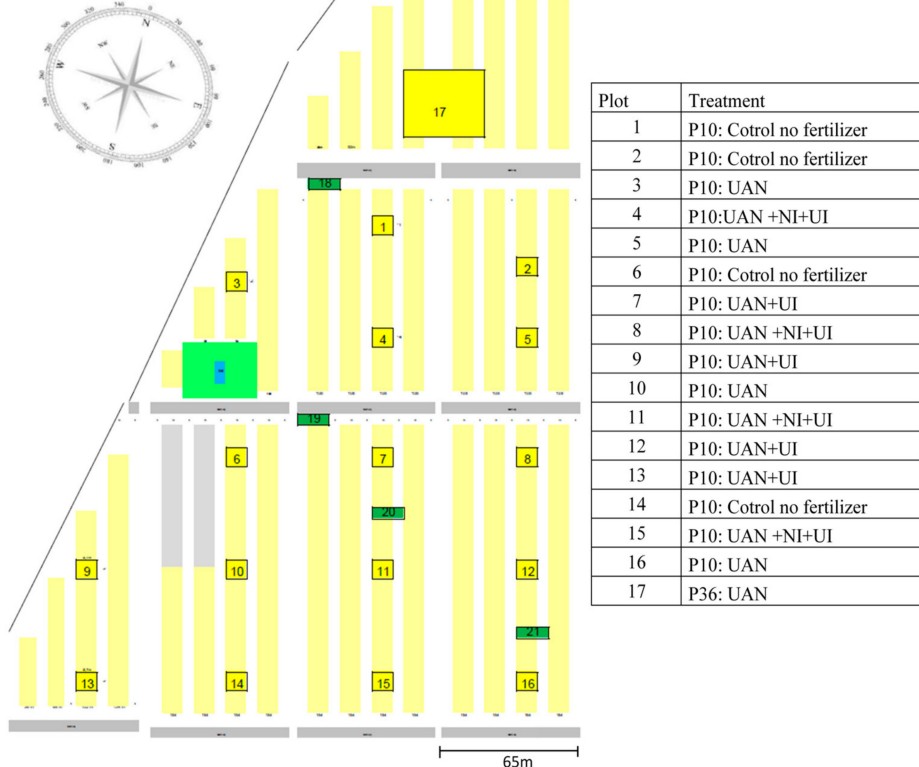

**Figure 1.** Location of plots and $NH_3$-ALPHA samplers (passive diffusion samplers measuring $NH_3$ concentration) for background measurements in March and April on the field in AArslev. The P10 plots ($10 \times 10$ m$^2$) are given the numbers 1–16 and the P36 plot ($36 \times 36$ m$^2$) number 17, i.e., plot used to measure $NH_3$ flux with passive flux samplers (Leuning samplers). The position of the ALPHA samplers giving background measurements are numbered 18–21. In Table 1 to the right of the figure, the treatment of each plot is given (NI = DMPSA, UI = NBPT).

## 2.2. Volatilization Measurements

Measurements of $NH_3$ emission were started immediately after application of UAN in 12 square plots with a side length of 10 m (P10) and one square plot with a side length of 36 m (P36) in experiment 1 (March 2018) and experiment 2 (April 2018), and after harvest to 9 square plots with a side length of 10 m in experiment 3 (August 2018). Background ammonia concentrations were measured upwind of the experimental plots and in the field between plots (Figure 1).

Ammonia emission from the P10 plots with UAN added and UAN with additives was determined using inverse dispersion modelling (IDM), which, in earlier articles [31–33], was named backward Lagrangian stochastic (bLS )modelling. The IDM model calculated $NH_3$ emissions from the plots using data from measurements of the wind speed using cup anemometers (3-cup anemometers coupled with Wind101A Data Loggers, MadgeTech, Inc., Warner, New Hampshire, NH, USA) and average $NH_3$ concentration, which was measured with four passive diffusion $NH_3$ samplers (ALPHA samplers) placed in the center of the plots in this study [27,34,35]. From the large P36 plot in experiment 1 and 2, the $NH_3$ emission was measured with the IDM method; $NH_3$ concentrations were measured with ALPHA samplers and passive flux samplers (Leuning samplers) and wind speed ($u$, m s$^{-1}$) measured with cup anemometers (MadgeTech, Inc., Warner, New Hampshire, NH, USA). In experiment 3, the passive flux samplers (Leuning samplers) were placed in a small P10 plot, where the $NH_3$ concentration was also measured with the ALPHA samplers. In all plots, the samplers were positioned in the center. Background $NH_3$ concentrations were provided with ALPHA samplers positioned at four different sites in the field and at heights reflecting the height of the measurements in the plots (Figure 1). The samplers in the plots were positioned at the $H_{zinst}$ height, which in the P10 plot was 0.5 m above

the soil surface in the study starting on 23 March (Exp. 1) and 16 August (Exp. 3), and 0.65 m above the soil surface (60 cm above crop canopy) when starting on 20 April (Exp.2). For the P36 plot, the $H_{zinst}$ height was 1.00 m [27].

The samplers were mounted in the field immediately after applying UAN to the plots in the morning and then shifted every morning at about 9:00. In all experiments, the samplers were exposed for 24 h in each measuring interval.

Measurement with ALPHA samplers is based on the principle of $NH_3$ diffusion through a PTFE membrane [34], with subsequent absorption to a filter paper coated with oxalic acid [36]. After exposure, absorbed $NH_3$ is leached using 4 mL of ultrapure water and quantified using an ammonium electrode (Orion High-Performance Ammonia Electrode, Thermo Fisher, PA). The mass of $NH_3$ absorbed is related to the gas phase concentration of the species by Fick's law of diffusion. Detailed procedures for handling ALPHA samplers and calculating the concentration of gaseous $NH_3$ are described in the work of [35]. The concentration ($\mu g$ $NH_3$-N m$^{-3}$) was related to the absorbed mass of $NH_3$ ($M$, $\mu g$ $NH_3$-N) by Fick's law of diffusion as follows:

$$\overline{NH_3} = \frac{M_{sample} - M_{control}}{V(t)}, \tag{1}$$

where $V$ (m$^3$) is the effective volume of sampled air, given by $V(t) = 0.0032 \times t$, where time $t$ is given in hours. During the trials, a set of samplers was placed in the center of each plot to measure the $NH_3$ concentration ($M_{sample}$) while the background or control concentration ($M_{control}$) was assessed by samplers placed west of the experimental plots.

The horizontal $NH_3$ flux ($F(x)$, $\mu g$ m$^{-2}$ s$^{-1}$) in the P36 plot was measured using passive flux samplers (Leuning samplers; [37]). This sampler is designed to give a measure of the mean horizontal flux density of $NH_3$ ($F(x)$,) which can be calculated using the following equation:

$$F(x) = \frac{M}{At}, \tag{2}$$

where $M$ is the mass of $NH_3$-N collected ($\mu g$ $NH_3$-N) by the oxalic acid coating on the interior of the passive flux sampler during the sampling period, $t$ (s), and $A$ is the effective cross-sectional area of the sampler (m$^2$). The average atmospheric $NH_3$ concentration (C, $\mu g$ m$^{-3}$) was then calculated as follows:

$$C = \frac{F(x)}{ut}. \tag{3}$$

The Leuning samplers were placed at $H_{zinst}$ height in duplicates in the center of the square plot. The preparation of the coated Leuning samplers followed the procedure outlined in the works of [37] and [38].

Ammonia emission from sources' surface areas was calculated using the inverse dispersion modelling (IDM) technique [31–33]. Volatilization rates were calculated by:

$$F_{IDM} = \frac{(C - C_b)}{\left(\frac{C}{F}\right)_{sim}}, \tag{4}$$

where $C$ and $C_b$ represent the $NH_3$ concentration at the point of measurement (PoM) and background, respectively ($\mu g$ m$^{-3}$), and $(C/F)_{sim}$ in (s m$^{-1}$) is the simulated ratio of the tracer concentration at PoM to the $NH_3$ emission rate $F_{IDM}$ ($\mu g$ m$^{-2}$ s$^{-1}$) from the source surface [32]. In this study, the atmospheric dispersion modelling software *WindTrax* (Thunder Beach Scientific, Halifax, Canada) was used to calculate $F_{IDM}$.

*2.3. Crop Response*

The yield crop response in the form of wheat biomass and N uptake was measured by harvesting half a meter of winter wheat with scissors at four different sites per plot on 17 April and 22 May 2018, and the plants' physiological age was determined using the BBHC scale [30] on 23 March, 17 April, and 22 May 2018. The BBHC scale is a uniform decimal code for the growth stages of crops and for cereals, i.e., EC00–EC10 is from seed to the emergence of coleoptile, EC11–EC20 is till 9 or more leaves unfolded, EC21–EC29 is till 9 or more sideshoots were visible, and EC30–EC39 is the maximum stem length reached for more stages [30]. On 25 July 2018, the wheat plants were harvested and grain yield was measured. The plant samples were dried at 80 °C for 24 h to provide the dry matter (DM) yield. The total plant N content was analyzed from the oven-dried material after burning the material at 900 °C, where the N-oxides and molecular N was determined by LECO TruSpec CN (St. Joseph, Michigan, MI, USA) as described in [39].

*2.4. Nitrate and Ammonium in Soil*

The amount of soil inorganic nitrogen (N) in the root zone of the 16 plots was measured by drilling out soil samples to 0.9 m with soil drills on 8 March 2018 before crop growth and fertilizer application and on 16 August after the harvest of winter wheat (Table 1). The soil sampling in August was carried out three weeks after harvest because it was impossible to drill in the dry soil at harvest, and soil sampling was instead carried out after rain had increased the soil water content. At each soil sampling, 10 cores were drilled per plot and were divided in three parts (0–30, 30–60, 60–90 cm). For each depth interval, the collected soil from 10 cores per plot was pooled for each depth interval, giving 3 composite samples from each plot. The amount of soil water in the soil samples was determined gravimetrically by drying at 80 °C for 20 h. Soil samples were frozen and sent to Agrolab (Sastedt, Germany) for an analysis of the nitrate ($NO_3^-$) and ammonium ($NH_4^+$). Immediately after thawing the soil samples, subsamples of 100 g of soil were added to 200 mL of KCL (1 M) and shaken for 1 h. After centrifugation, the supernatant was analyzed for the $NO_3^-$ and $NH_4^+$ concentration with continuous flow injection analysis by an AutoAnalyzer 3 (Bran+Luebbe GmbH, Norderstedt, Germany) see [40].

*2.5. Data Analysis*

Statistical analyses were performed using SAS JMP 13 [41]. Differences between treatment means at $\alpha = 0.05$ were assessed with one-way analysis of variance (ANOVA) applying the Tukey post hoc test (SAS 2013).

## 3. Result

The air temperature during the study ranged between −5 and 25 °C (Figure 2). There were no rain events during the first 76 h of the three experiments. In experiment one, the temperature varied between −5 and 7 °C during the first 225 h, increasing to between 2 and 25 °C in experiment two and between 10 and 25 °C in the third experiment.

In experiment one, it rained with an intensity of up to 4 mm h$^{-1}$ 76 to 82 h after application of UAN, and in experiment two, a heavy rain event occurred after 112 h, whereas in the experiment, there was a short period with low intensity rain from 110 to 115 h after UAN application. In the first experimental period, the wind speed measured at 10 m ranged between 0 and 8.3 m s$^{-1}$. The wind speed varied between 0 and 12 m s$^{-1}$ in the second experiment and was extremely unstable. In experiment 3, the wind speed varied between 2 and 7 m s$^{-1}$.

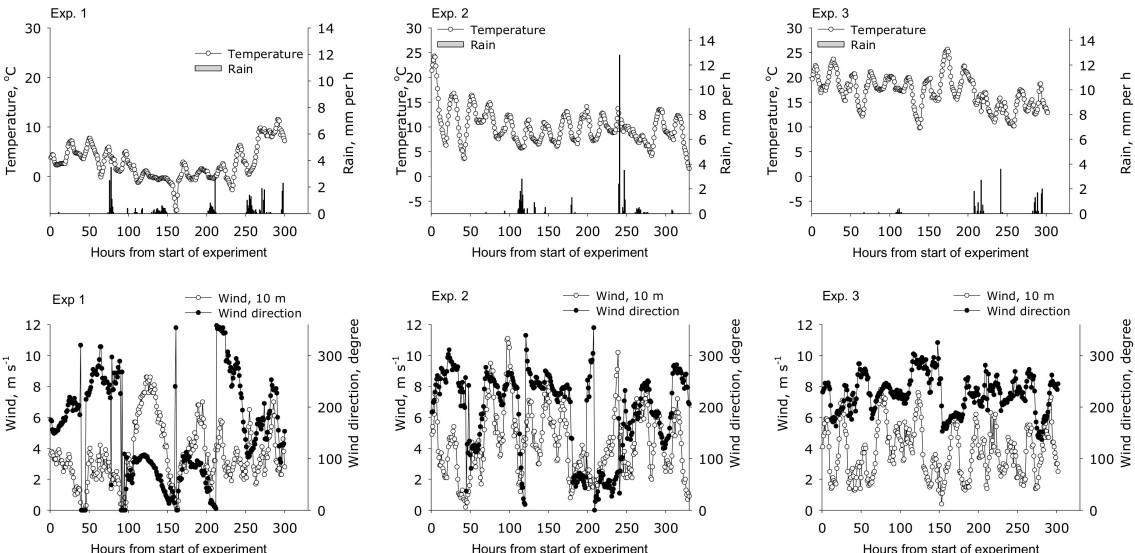

**Figure 2.** Weather conditions during the experiment in experiment one (March), experiment two (April), and experiment three (August). Top: Temperature and rain. Bottom: wind speed and wind direction at 10 m height (source: Danish Meteorological Institute).

There were no prevailing wind directions, reflecting the very unstable weather during the two first experimental periods. In experiment three, the wind direction was from the south and west, the direction from which there were no hedgerows.

From the beginning of May until August, it did not rain, and the temperature was higher than 20 °C during daytime and higher than 15 °C during nighttime.

*3.1. Ammonia Emission*

In experiment one (March), where UAN was added to plots with seedlings (BBCH 10–12, first leaf unfolded), the $NH_3$ concentration in 825 of the samples (87.12%) was below the limit of detection (LOD) for the ammonia selective electrode, which was 5 $\mu$M, and the $NH_3$ emission was set to zero for the time intervals represented by these data [36]. There were no significant differences in the emissions of $NH_3$ between the UAN treatments in this experiment ($p > 5\%$), with cumulative $NH_3$ emissions of less than 2% of the N applied.

In experiment two (April), where plants were at BBHC growth stage 15 (five leaves unfolded), the background $NH_3$ concentrations were high and the P36 plot was situated in a site where $NH_3$ concentrations were not significantly different from the background $NH_3$ concentration measured in this part of the field, and no significant emission of $NH_3$ was measured from the large plot. Using the background concentration data from samplers positioned upwind, the P10 plots treated with urea provided valid $NH_3$ emission data, showing that after fertilizing with UAN in experiment two (April), the rate of $NH_3$ emission from all plots increased up to 70 h after the initiation of the study. Thereafter, $NH_3$ emissions decreased during rain events, and then increased and were high till 170 h after the start. Hereafter, $NH_3$ emissions were negligible until between 200 and 250 h, and increased after 250 h to between 0 and 0.02 kg N ha$^{-1}$ h$^{-1}$. After 350 h, emissions were insignificant (Figure 3).

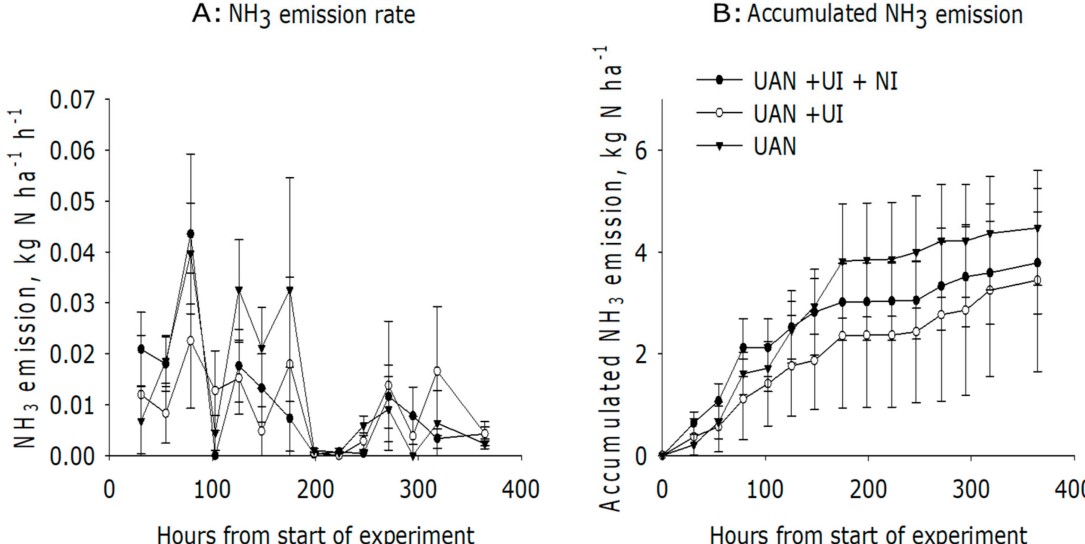

**Figure 3.** Experiment two where NH$_3$ emission was measured in April. A: NH$_3$ emission rates. B: accumulated NH$_3$ emission (B). The treatments were UAN; UAN without additives, UAN + UI; UAN mixed with urease inhibitor (UI) or UAN + UI + NI; UAN mixed with UI and nitrification inhibitor (NI). (Error bars: SE, *n* = 4).

This pattern is reflected in the accumulated emission of NH$_3$, which, in experiment two, increased until 170 h after the start of the experiments, but after 170 h, little increase in NH$_3$ emissions was measured. There was no significant difference in the NH$_3$ emission rates (*p* > 5%) between treatments. Further, NH$_3$ emission was significant higher in this study compared to NH$_3$ emissions measured in March (Exp. 1).

In experiment 3, after harvest, the NH$_3$ emission rate from the plot with added UAN measured with Leuning samplers was initially lower than from the plots where NH$_3$ emission was measured with ALPHA samplers (Figure 4), but the NH$_3$ emission from this plot increased after the first 24 h and the cumulated NH$_3$ emission measured with the two measuring techniques (Figure 4) was not significantly different (*p* < 5%). In this experiment, the background NH$_3$ concentration was low and the coefficient of variation of the average background NH$_3$ concentration was less than 3%. In the first measuring period, the sampling and measurements of the NH$_3$ concentration from plot 1 and plot 8 failed, and it was decided that an average of the two NH$_3$ emission measurements from two plots would be used, given the same treatments and measured in the same time interval. An analysis of the data showed there were no significant differences in the NH$_3$ emission rates between these (*p* > 5%). This allowed us to calculate the standard error of cumulated NH$_3$ emissions and the coefficient of variation of the NH$_3$ emissions from the three treatments using data from all plots (Figure 5).

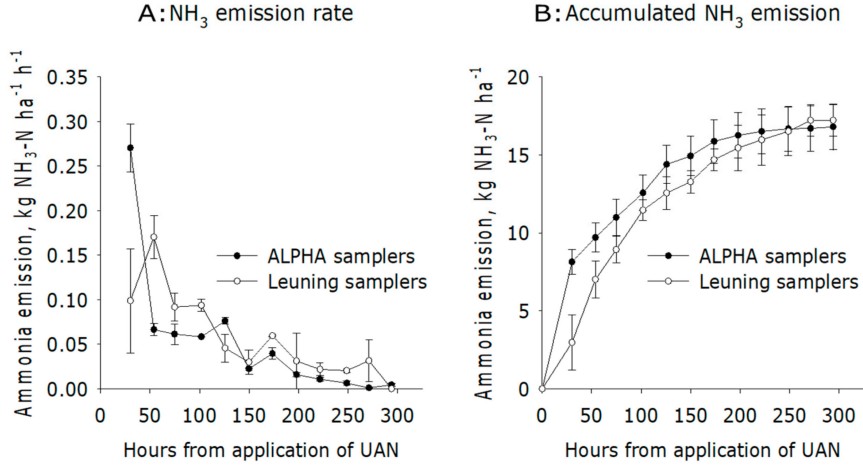

**Figure 4.** Test of the measuring technique in August, where the rate of $NH_3$ emission from a urea ammonium nitrate (UAN)-treated plot was measured with passive flux samplers (Leuning samplers) and passive concentration samplers (ALPHA samplers). A: $NH_3$ emission rate. B: cumulative $NH_3$ emission (Error bars: SE, $n = 3$).

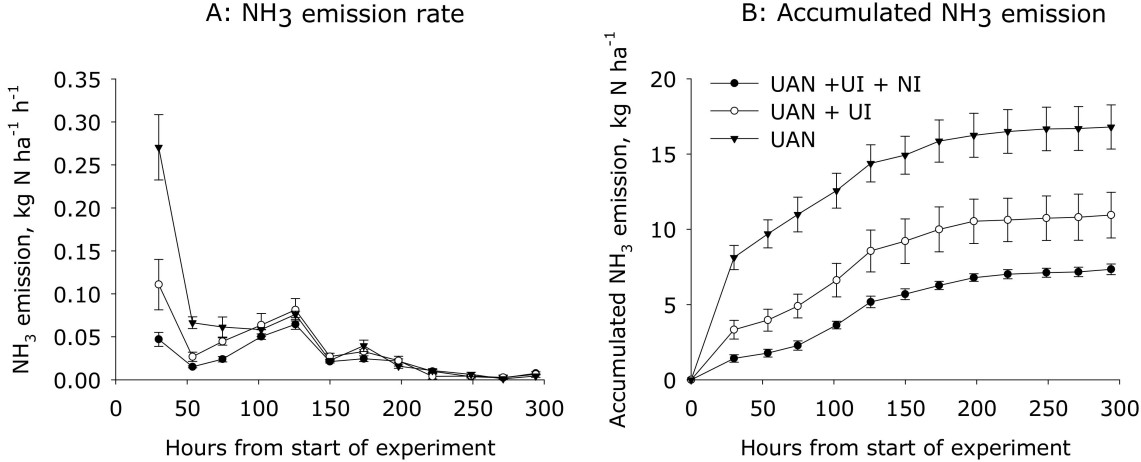

**Figure 5.** Experiment three where $NH_3$ emission was measured in August. A: $NH_3$ emission rate. B: accumulated $NH_3$ emission. The treatments were UAN; UAN without additives, UAN + UI; UAN mixed with urease inhibitor (UI) or UAN + UI + NI; UAN mixed with UI and nitrification inhibitor (NI) (Error bars: SE, $n = 3$).

In August, the $NH_3$ emissions from fallow soil were high for the first 24 h after application of UAN, and after 24 h, $NH_3$ emissions declined. The emissions increased after 75 h, and thereafter, they decreased and were negligible after 200 h (Figure 5). During the first 100 h, the $NH_3$ emission from the plots with added UAN was higher than from the plots with added UAN mixed with additives ($p < 5\%$), and the accumulated $NH_3$ emission from the plots with added UAN mixed with UI were not significantly higher ($p < 5\%$) than from the plots with added UAN mixed with UI and NI (Figure 6).

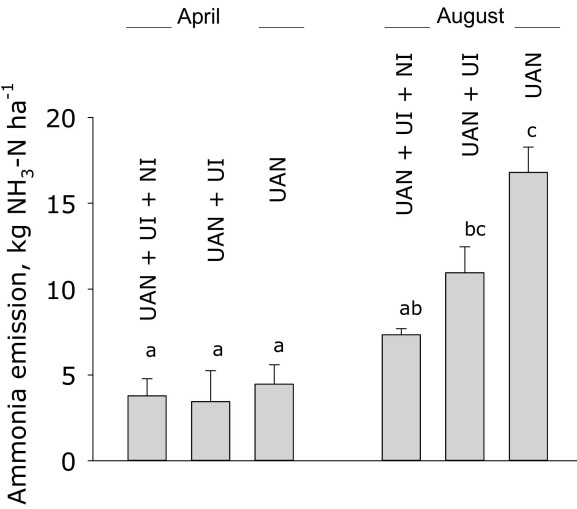

**Figure 6.** Accumulated emissions of NH$_3$ from the plots after the application of UAN in April (Error bars: SE, *n* = 4) and August. The treatments were UAN; UAN without additives, UAN + UI; UAN mixed with urease inhibitor (UI) or UAN + UI + NI; UAN mixed with UI and nitrification inhibitor (NI) (Error bars: SE, *n* = 3).

While NH$_3$ emissions were negligible in experiment one, in experiment two, the cumulated NH$_3$ emissions were 2.5% of the N added in UAN + UI + NI, 2.7% of the N in UAN + NI, and 3.1% of the N in UAN. The cumulated NH$_3$ emissions in experiment three from the plot with added UAN were 17% of the N, 11% from the UAN + UI, and 7% from the UAN + UI + NI.

### 3.2. Plant Growth and Harvest Yield

The physiological stages of the wheat plants were EC 10–12 on 23 March, EC 15 on 17 April, and EC 37–41 on 22 May (BBHC scale, Table 1), indicating that growth started late in spring 2018. In May, biomass production in the UAN-fertilized plots was much higher (*p* < 5%) than in the control, which was not the case in April (Figure 7). At harvest, the grain yield in the plots where the UAN solution was supplemented with additives was higher than in the unfertilized plot (*p* < 5%), but there were no significant differences in the yield between the three fertilizer treatments, though a trend of higher yields in UAN treatments with inhibitors was identified, i.e., the addition of inhibitors increased the grain yield with 7% (UAN + UI + NI) and 14% (UAN + UI).

N uptake between sampling dates and treatments was similar to the yield data. Though not significant, both treatments with inhibitor showed higher N uptake at a similar magnitude to the observed yield effects.

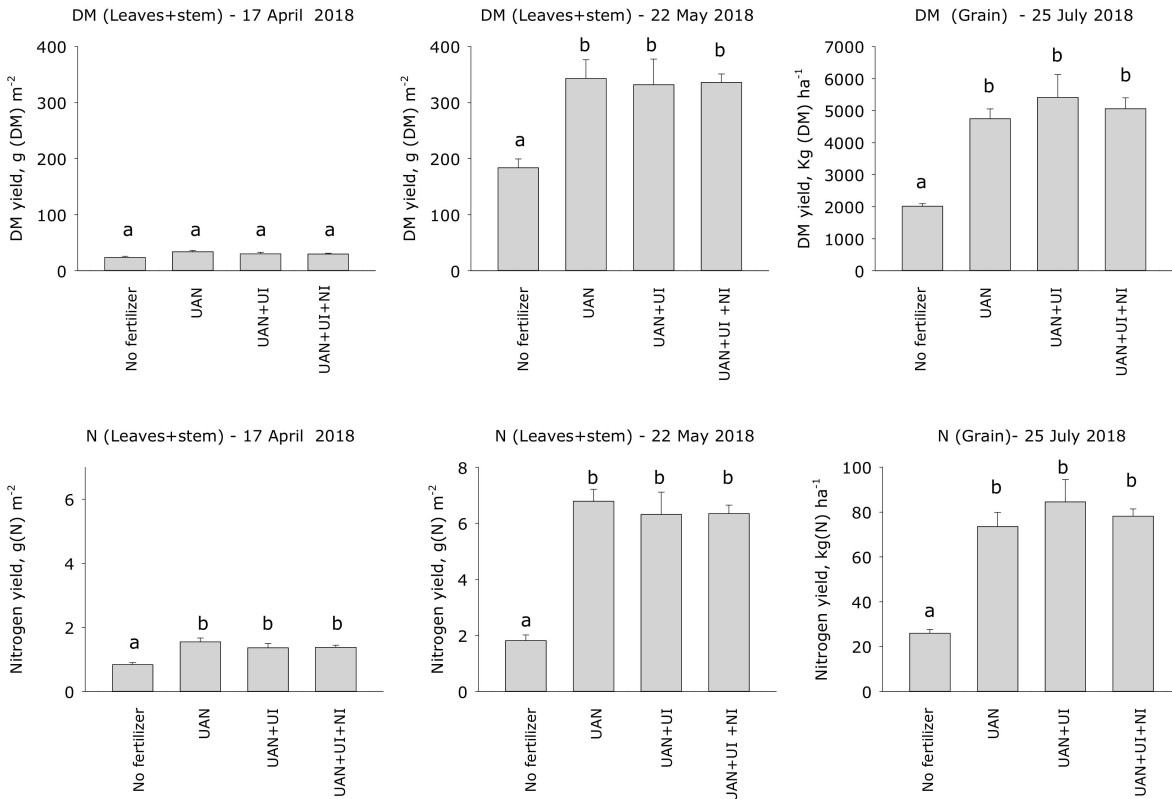

**Figure 7.** Winter wheat plant leaves and stem on 17 April and 22 May 2018 and harvested grain yield on 25 July 2018. Top dry matter yield and bottom N yield. The treatments were UAN; UAN without additives, UAN + UI; UAN mixed with urease inhibitor (UI) or UAN + UI + NI; UAN mixed with UI and nitrification inhibitor (NI) (Error bars: SE, *n* = 3).

### 3.3. N in Soil

In March, before the application of UAN, all plots had a similar content of inorganic N in the soil profile from the 0–90 cm depth (Figure 8). The inorganic N amount varied from 23 to 28 kg N ha$^{-1}$ and the ratio of $NH_4$ to $NO_3$ was 1 to 4. Due to the drought during summer 2018, soil samples could not be drilled immediately after harvest but were sampled on 16 August 2018 after rain events, i.e., 3 weeks after harvest. The inorganic N content of the unfertilized soil was about two times higher in August than in March, and in the fertilized plots, the content was between 156 and 206 kg N ha$^{-1}$, the lowest level in the plots treated with UAN + UI and the highest levels in plots with added UAN + UI + NI. The content of $NH_4^+$-N in the soil was higher in the UAN-treated plot than in the untreated plot, and much higher in the UAN + UI + NI treatment than in the UAN and UAN + UI plot. In these plots, 75% of the inorganic N was found in the layers from 0 to 30 cm.

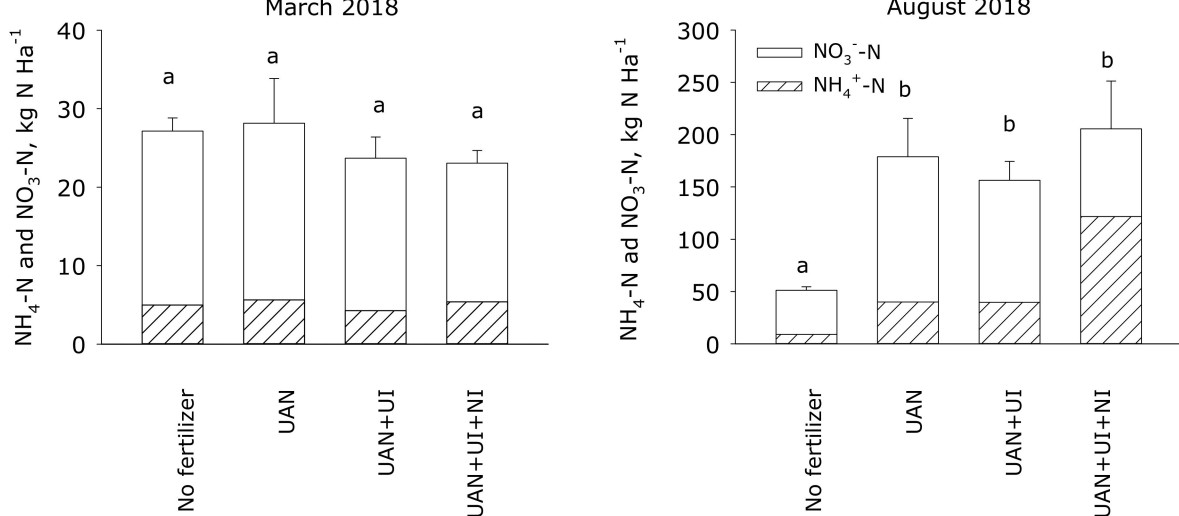

**Figure 8.** Content of $NH_4$-N and $NO_3$-N in the soil profile from 0 to 90 cm in March (left) and August (right) 2018. The treatments were UAN; UAN without additives, UAN + UI; UAN mixed with urease inhibitor (UI) or UAN + UI + NI; UAN mixed with UI and nitrification inhibitor (NI) (Error bars: SE, *n* = 3).

## 4. Discussion

The current study was performed for one year, with the aim of measuring $NH_3$ emissions as affected by plant height and relating this emission to crop yield. Crop yield and $NH_3$ emission are affected considerably by climate, but this was not accounted for in this study.

Due to low temperatures combined with high amounts of precipitation shortly after the first fertilizer application in experiment one, $NH_3$ emissions were reduced to a negligible level, which is consistent with previous studies [2]. Under such weather conditions, urea-derived $NH_3$ emissions are very low and emission reduction with inhibited urea is not required.

The high $NH_3$ background concentration in experiment two was probably due to $NH_3$ emission from the urea containing NPK fertilizer, which was applied on 14 April in the field surrounding the P10 plots to which liquid urea was added 6 days later on 20 April. As a result of high background concentrations of $NH_3$, the measured $NH_3$ emissions from the UAN plots may have been low in experiment two compared to many studies of $NH_3$ emission from urea applied in the field [3] but in the same range found in the study of [15]. In addition to the high background $NH_3$ concentration found for rain events after urea application [16], lower air temperatures and crop height also contributed to the low $NH_3$ emission levels [42]. The emissions from plots with UAN added with inhibitors were reduced by 13% to 20% (not significant) compared to emissions from plots with UAN added. The reduction at this occasion was low compared to recent studies showing reductions of 39% to 50% [43,44].

In experiment three, the cumulative $NH_3$ emissions from UAN were 17% of the applied N, which is higher than the European $NH_3$ emission factor of about 10% to 12% [8] but lower than that found in studies using wind tunnels [14]. Studies have shown that NIs may increase $NH_3$ emissions from urea [22] or urine [45], but in the presented study, $NH_3$ emissions were reduced by 35% through the addition of UI and 60% by UI with NI. This reduction is in line with results showing that UI and the combination of UI and NI may reduce $NH_3$ emissions from urea by 39% to 50% [43,44]. Objective two and three are partly supported by the presented study due to the negligible $NH_3$ emissions in experiment one and the effect of high background concentrations in experiment two, but in experiment three, a similar reduction of $NH_3$ emissions by UI in UAN to the emission from urea was shown. In addition, the results of experiment three also support hypothesis 3 as the UI + NI treatment gave the same and an even higher $NH_3$ emission reduction compared to UI only.

The new approach of multiple small plot measurements of $NH_3$ emissions was proved feasible by yielding $NH_3$ emissions close to those observed in other studies with standard methods documented in the EMEP Guidebook for UAN and by a comparison with simultaneously measured $NH_3$ emissions by standard large-scale sampling using Leuning-Samplers. So, the first hypothesis of our study is supported by the data. A similar approach was also applied with success by [46] while another multi-plot method with acid trap passive flux samplers also allowed differentiation of treatment effects [26,47]. As a result, standards for multi-plot $NH_3$ measurements with passive flux samplers should be developed by the groups that have successfully applied this approach. The second application yielded somewhat biased results due to the high background concentrations. However, this was a trial site management artefact due to an incorrect selection of the fertilizer source for the surrounding guard areas, which can be easily avoided in future experiments.

Due to a cold, wet, and dark spring, plant growth was low in March and April, and the drought from May till August meant that the yield was 20% lower than that seen in normal years. The addition of inhibitors increased the grain yield by 7% (UAN + UI + NI) and 14% (UAN + UI), which is similar to the increase in corn yield at 5% to 7% due to the addition of UI as measured by [14]. In line with increased yields, N uptake was stimulated by the application of the inhibitors, showing a positive effect on N use efficiency, and supporting our fourth hypothesis. Similar effects were reported in recent review studies [48]. However, yield and N uptake effects were not statistically significant, although average effects were high. Variation within treatments was high (Figure 7), probably due to the micro-site water supply effect typical for drought years, i.e., the variability of soil characteristics affects the yield more in dry years than in years with sufficient water supply. Nevertheless, the data support a yield increasing effect of the added inhibitors, which needs to be further corroborated under better suited experimental conditions.

The high concentration of inorganic N in the plots supplied with UAN indicates that a substantial part of the added fertilizer N was not taken up by the wheat crop, which gave a comparatively low yield. In addition, the mineralization of organic N may be high after a rain event following a drought, and the soils at the Aarslev research station are known to be fertile. The organic N pool in the soil may have contributed to the high inorganic N content measured 3 weeks after harvest and after rain increased the soil water content and increased mineralization. Therefore, a large share of the high post-harvest $N_{min}$ values probably stemmed from mineralized organic N and not from residual mineral N applied with the fertilizers, which makes a comparison of the treatment effects difficult. Nevertheless, the lower level of inorganic N in the soil from plots treated with UAN + UI compared to plots with UAN and UAN + UI + NI may be explained by the higher harvest of grain in the plots given this treatment. The high $NH_4$ content in the soil with UAN and NI added shows the long-lasting effect of reduced nitrification of $NH_4$ in the soil due to the applied NI.

## 5. Conclusions

A new multi-plot measurement approach was successfully applied on a small plot basis to obtain quantitative $NH_3$ losses from a field with applied UAN, and UAN treated with UI and a mixture of UI and NI. Low and biased values in one of the three experiments highlight the importance of controlled experimental conditions with low $NH_3$ background concentrations, which requires high awareness in future experimentation.

Emission levels of UAN-derived $NH_3$ varied between applications, with the highest $NH_3$ emissions found in late summer (17% of applied N).

Yields non-significantly increased by between 7% and 14% due to the addition of inhibitors, due to the drought causing large variation in the average yields of the treatments. The results prove the reduction effect of the added inhibitors on $NH_3$ emissions, which was translated as higher yields and N uptake. Future research is needed to corroborate the observed effects and to develop unified protocols for multi-plot measurement of $NH_3$ emissions with passive flux samplers/ALPHA samplers and micro-meteorological IDM modelling.

**Author Contributions:** M.T.N., S.G.S. and A.S.P. jointly developed the aim of the project, M.T.N. and S.G.S. designed the experimental lay out, M.T.N. carried out the field experiment and the data analysis, M.T.N. and S.G.S. wrote the first draft of the article and all authors contributed to the final version of the manuscript. All authors have read and agreed to the published version of the manuscript.

**Acknowledgments:** The authors would like to thank Astrid Bergmann, Lasse Lillevang Vesterholt and Jens Rasmus Dalgaard Elkjær Aarhus University for their technical assistance, and Jakob Kure, Jakob Krabben, Brian Jønson and Lars Duelund from University of Southern Denmark who provided valuable support in carrying out work in the field and laboratories.

**Conflicts of Interest:** The authors declare no conflict of interest.

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
