# Peer review of "Urea Ammonium Nitrate Solution Treated with Inhibitor Technology: Effects on Ammonia Emission Reduction, Wheat Yield, and Inorganic N in Soil"

_agronomy, doi:10.3390/agronomy10020161_

Round 1
Reviewer 1 Report
The manuscript entitled “Urea Ammonium Nitrate Solution treated with Inhibitor Technology: Effects on Ammonia Emission Reduction, Wheat Yield and Inorganic N in Soil” is focusing on a topic that is both relevant and important for the sustainable management of wheat cropping systems. Greenhouses gas emissions (GHGs) is a current challenge in the agricultural field and understanding the mechanisms which can reduce agricultural GHGs production is a key step for agriculture sustainability. However, the manuscript and the experiment itself miss some key features.
In particular, the major concern is that the experiment was performed only one year which limits the possibility of a robust data discussion and conclusion. A field experiment should be performed for at least two years.
The hypotheses are not clear and difficult to understand; especially points 2) and 3).
Material and methods should provide more details about the experimental design and the applied fertilizers (also about P and K). In addition, in the data analysis section, more details about the statistical analysis should be provided.
Finally, the manuscript should provide the reason for the agronomical technics applied; for instance why the application of UAN after the harvest.
Author Response
In particular, the major concern is that the experiment was performed only one year which limits the possibility of a robust data discussion and conclusion. A field experiment should be performed for at least two years.
Response. Thank you this is a valid point – the text has been amended to include this point as follows: “The current study was performed for one year with the aim to measure ammonia emission as affected by plant height and relate this emission for crop yield. Crop yield and ammonia emission is much affected by climate, but this is not accounted for in this study.
The hypotheses are not clear and difficult to understand; especially points 2) and 3).
Material and methods should provide more details about the experimental design and the applied fertilizers (also about P and K). In addition, in the data analysis section, more details about the statistical analysis should be provided.
Response:Thank you
The sentence presenting hypothesis and objectives has been reworded as follows:“In the study the intention was to prove the hypothesis objectives were to test: 1) that a multiplot micrometeorological method provide valide NH3 emission measurement, and 2) test that treatment of UAN with UI reduces NH3 emissions to aas efficient as similar extent to a level similar with the effect when treatingtreatment of urea with UI, 3) test a combination of UAN with UI and NI reduces NH3 emissions as the UI treatment as in studies with urea and 4) the reduction of NH3 emissions and NH4+ stablilization with the two inhibitors provide high crop yield and affect N species (NO3-, NH4+) distribution in soil.”
More information about P and K addition to the plots is provided in the table 1.
Finally, the manuscript should provide the reason for the agronomical technics applied; for instance why the application of UAN after the harvest.
Reponse: UAN is applied after harvest to simulate application of N to grassland after winter wheat.
Reviewer 2 Report
Reviewed work represents a new approach to the problem of the low efficiency of nitrogen fertilization, mainly due to the application of nitrogen in urea form. The work is very interesting because it explores a new methodology to measure nitrogen losses in ammonia form, and the effect of the use of different inhibitors.
Since the description of the methodology contains some punctuation, grammar and syntax errors, it makes it hard for the reader to fully understand it. Then, I recommend reviewing my suggestions to be able to continue revising the obtained results in depth.
I would appreciate if the authors could make a careful reading of the manuscript, since there are some mistakes that are probably typos.
In all the manuscript, revise double spacing between words because this aspect needs more attention, i.e., in lines 40, 43, 27, 48, 57, 60, 65, 138, 223, 351 and 359.
Lines: 19, 74, 96: ¿Is it the correct way to write the molecule name? I think that it is 3,4-Dimethylpyrazole, unless it is referring to another molecule.
Line 25: … additives was observed (because it is understood that it was measured and the effect could be observed or not).
Line 96: Within the parenthesis, you could use the abbreviation DMPSA established before.
Line 30: The keywords ammonia and emission should go together because you only measured the emissions of this gas.
Line 38: Space urea·is
Line 39: All the verbs in this sentence should be in the third singular person form: increases, shifts, causes, results. Please review grammar in all manuscript.
Line 40: The correct percentage is 64%.
Line 42: It is unclear what PM2.5 means the way it is written. On the one hand, 2.5 it should be written as subscript, and on the other hand, it is recommended to write it more clearly, for instance, “forming particulate matter as fine aerosol (PM2.5).
Line 46, 55, 388: Use the abbreviation established before (UI). In the line 412 use only “UI” or “urease inhibitor”.
Line 56: Its efficiency is… or UI efficiency is…
Line 58: The sentence “when application…” is unclear.
Line 60: “provide reduce emission” is unclear. Furthermore, it should be “NH3 emission”. Review this in all manuscript.
Line 62: Since it is the first time you mention N2O, you should write its full name: nitrous oxide (N2O).
Lines 74, 96, 366: Use the previously established abbreviation (NI).
Line 69: all losses pathways
Line 82: … the gas emissions…
Line 85: Correct “micrometeoroogical”
Line 96: Use the abbreviation DMPSA
Line 97: “the effect of no”?
Line 98: Are you announcing hypotheses or objectives? Because the first is written as an objective but the rest are written as hypotheses. The objective or hypothesis number 2 is unclear.
Line 114: Mention the distance from that station to the experimental site.
Table 1: What is “Sherif”? If it is a winter wheat variety, indicate (var.), if not, please, clarify. If it is a variety, the correct spelling is Sheriff. Consider this too in the line 20.
Fig 1: Neither the numbers close to the yellow bars nor the numbers included in the grey zones are visible. It is more clear if you write “NH3 ALPHA samplers”.
Line 124: “vulgare” should go in undercase. This paragraph should be reviewed. Winter wheat was drilled in August or in September (as it is written in Table 1)?
Line 128: “EC” is a growth stage. But specify it in the manuscript and include a reference.
Line 132: Table in uppercase.
Line 180: fig. appears sometimes as Fig.
Table 1: what is the meaning of the asteriscs in the treatments in the third column? In the second column, fourth row, there are two stops “S..” It is recommended to write titles in the table. Furthermore, I suggest reviewing this table because it has too much information and many aspects, such as doses of nitrogen applied in each treatment, are not easy to understand. I think that two tables, one with a timeline and other with the relevant aspects of each treatment, would be more suitable.
Line 140: I think that the term “campaigns” is not correct because there are not three different crop cycles, maybe it is better to use only “three experiments”.
Lines 140-144: It is the same information as in Table 1.
Line 145: A space is needed between “…-1” and “liquid”.
Line 159: What amount of P and K? Did this amount vary among plots? In M&M, I miss some chemical soil characteristics such as organic matter, P, K, and total N contents.
Line 162-166: Given that each plot is numbered in Figure 1, it would be interesting to describe which were the ones measured to know which treatment they belong to.
Equation 1: I do not understand the meaning of the upper line in NH3.
Equations 2, 3 and 4: Consider reviewing the font size since it is bigger than in the first equation.
Line 200: Something is missing after “which is”.
Line 204: Write “(C)” after “NH3 concentration”.
Line 214: A parenthesis is missing.
Line 218: “and-the”
Line 236: “from from”
Line 237: As it is written, it is not clear if it was the soil or the soil extract that was frozen.
Line 250: If the value can be negative, the separation with a dash can be confusing. Please use “and” always, as in the line 248.
Figure 2: Change the semicolon to colon after “top” and “bottom”.
Section 3.1: Sometimes you define the stages of your work as “experiment” and some other times you do it as dates. Given the complexity of your work, I think it is more understandable if you only use one criterion.
Fig 5: The quality and edition of this figure is worse than in the previous ones.
Figures 3, 4 and 5 are composed by two graphs, they could be numbered or identified with letters. In figures 3 and 5, the graph on the right represents the same information as in figure 6 (accumulated emissions of NH3). Maybe the better figure to represent the accumulated NH3 emissions is figure 6.
Figure 7: The vertical title of the first graph in not visible.
Use the same criterion when placing the statistics letters: close to the error bars, lined up on top or lined at the bottom of the bars.
Figure 8: Correct the abbreviation of ammonia and nitrate in the vertical title (symbols + and –).
Section 3.3: I am missing the results of nitrogen content at the different depths sampled.
Line 364: Could you please explain why? You are using a new methodology and it could be interesting to develop your results in this aspect.
Author Response
In all the manuscript, revise double spacing between words because this aspect needs more attention, i.e., in lines 40, 43, 27, 48, 57, 60, 65, 138, 223, 351 and 359.
Response: The typos have been corrected
Lines: 19, 74, 96: ¿Is it the correct way to write the molecule name? I think that it is 3,4-Dimethylpyrazole, unless it is referring to another molecule.
Response: Thank you we have now presented the correct chemical formula of the molecule.
Line 25: … additives was observed (because it is understood that it was measured and the effect could be observed or not).
Thank you, text is corrected by omitting “measured” and including “observed”.
Line 96: Within the parenthesis, you could use the abbreviation DMPSA established before.
Response: Thank you we are now writing DMPSA
Line 30: The keywords ammonia and emission should go together because you only measured the emissions of this gas.
Response: Thank you this is now corrected.
Line 38: Space urea·is
Response: Thank you this typo is corrected.
Line 39: All the verbs in this sentence should be in the third singular person form: increases, shifts, causes, results. Please review grammar in all manuscript.
Response: Thank you the grammar is corrected
Line 40: The correct percentage is 64%.
Response: Thank you this error has been corrected.
Line 42: It is unclear what PM2.5 means the way it is written. On the one hand, 2.5 it should be written as subscript, and on the other hand, it is recommended to write it more clearly, for instance, “forming particulate matter as fine aerosol (PM2.5).
Response: Thank you we have the text is corrected as proposed by reviewer.
Line 46, 55, 388: Use the abbreviation established before (UI). In the line 412 use only “UI” or “urease inhibitor”.
Response: Thank you we have the text is corrected as proposed by reviewer
Line 56: Its efficiency is… or UI efficiency is
Response: Thank you, reworded as follows “UI efficiency is”
Line 58: The sentence “when application…” is unclear.
Response: Thank you, reworded as follows “if rainfall follows within few days after application of urea”
Line 60: “provide reduce emission” is unclear. Furthermore, it should be “NH3 emission”. Review this in all manuscript.
Response: Thank you, throughout the text we write “NH3 emission” when appropriate.
Line 62: Since it is the first time you mention N2O, you should write its full name: nitrous oxide (N2O).
Response: Thank you we have included nitrous oxide.
Lines 74, 96, 366: Use the previously established abbreviation (NI).
Response: Nitrification inhibitors has been replaced with NI.
Line 69: all losses pathways
Response: Thank you the sentence has been reworded.
Line 82: … the gas emissions…
Response: The sentence has been corrected.
Line 85: Correct “micrometeoroogical”
Response: Spelling has been corrected.
Line 96: Use the abbreviation DMPSA
Response: Thank you the abbreviation is used.
Line 97: “the effect of no”?
Response: Thank you “no crop”
Line 98: Are you announcing hypotheses or objectives? Because the first is written as an objective but the rest are written as hypotheses. The objective or hypothesis number 2 is unclear.
Response: the sentence has been reworded as follows:“In the study the intention was to prove the hypothesis objectives were to test: 1) that a multiplot micrometeorological method provide valide NH3 emission measurement, and 2) test that treatment of UAN with UI reduces NH3 emissions to aas efficient as similar extent to a level similar with the effect when treatingtreatment of urea with UI, 3) test a combination of UAN with UI and NI reduces NH3 emissions as the UI treatment as in studies with urea and 4) the reduction of NH3 emissions and NH4+ stablilization with the two inhibitors provide high crop yield and affect N species (NO3-, NH4+) distribution in soil.”
Line 114: Mention the distance from that station to the experimental site.
Response: Thank you the distance was 750 m – this is now mentioned.
Table 1: What is “Sherif”? If it is a winter wheat variety, indicate (var.), if not, please, clarify. If it is a variety, the correct spelling is Sheriff. Consider this too in the line 20.
Response Sheriff is a variety of wheat we are now writing ‘Sheriff’ (var.)
Fig 1: Neither the numbers close to the yellow bars nor the numbers included in the grey zones are visible. It is more clear if you write “NH3 ALPHA samplers”.
Response: Thank you we are now writing NH3-ALPHA samplers and the size of the figure has been increased.
Line 124: “vulgare” should go in undercase. This paragraph should be reviewed. Winter wheat was drilled in August or in September (as it is written in Table 1)?
Response: Thank you the text has been reworded.
Line 128: “EC” is a growth stage. But specify it in the manuscript and include a reference.
Response: Thank you we are now mentioning that EC is a growth stage and that we use the BBHC and refer to Lancashire, P.D.; Bleiholder, H.; Van De Boom, T.; Langelüddeke, P.; Stauss, T.; Weber, E.; Witzenberger, A. A uniform decimal code for growth stages of crops and weeds. Ann. appl. Biol. 1991, 119, 561-601. |https://doi.org/10.1111/j.1744-7348.1991.tb04895.x,
Line 132: Table in uppercase.
Response: Thank you Table is now with in uppercase T.
Line 180: fig. appears sometimes as Fig.
Response: Thank you Fig. is now with and uppercase F.
Table 1: what is the meaning of the asteriscs in the treatments in the third column? In the second column, fourth row, there are two stops “S..” It is recommended to write titles in the table. Furthermore, I suggest reviewing this table because it has too much information and many aspects, such as doses of nitrogen applied in each treatment, are not easy to understand. I think that two tables, one with a timeline and other with the relevant aspects of each treatment, would be more suitable.
Response: Thank you very much for the recommendations, the table has been rewised according to the proposal given.
Line 140: I think that the term “campaigns” is not correct because there are not three different crop cycles, maybe it is better to use only “three experiments”.
Response Thank you we have omitted campaigns.
Lines 140-144: It is the same information as in Table 1.
Response: I think this information contribute to clarify the setup of the experiments.
Line 145: A space is needed between “…-1” and “liquid”.
Response: Thank a space is now included
Line 159: What amount of P and K? Did this amount vary among plots? In M&M, I miss some chemical soil characteristics such as organic matter, P, K, and total N contents.
Response: The amount of P and K did not vary among plots. This is now clarified in the table.
Line 162-166: Given that each plot is numbered in Figure 1, it would be interesting to describe which were the ones measured to know which treatment they belong to.
Response: Thank you this is mentioned in fig. 1.
Equation 1: I do not understand the meaning of the upper line in NH3.
Response: In the line above the equation is mentioned that M is the amount of NH3 sampled.
Equations 2, 3 and 4: Consider reviewing the font size since it is bigger than in the first equation.
Response: Thank you all equation has been reformatted.
Line 200: Something is missing after “which is”.
Response: Thank you “which is” is deleted.
Line 204: Write “(C)” after “NH3 concentration”.
Response: Thank you “C and units” are now mentioned.
Line 214: A parenthesis is missing.
Response: Thank you the parenthesis “)” is now included.
Line 218: “and-the”
Response: Thank you “-” is deleted.
Line 236: “from from”
Response: Thank you “from” is deleted.
Line 237: As it is written, it is not clear if it was the soil or the soil extract that was frozen.
Response: Thank you the sentence has been reworded to “Soil samples was frozen and send….”
Line 250: If the value can be negative, the separation with a dash can be confusing. Please use “and” always, as in the line 248.
Response: Thank you we are now writing “and”
Figure 2: Change the semicolon to colon after “top” and “bottom”.
Response: Thank you semicolon is changed to colon.
Section 3.1: Sometimes you define the stages of your work as “experiment” and some other times you do it as dates. Given the complexity of your work, I think it is more understandable if you only use one criterion.
Response: Thank you we have now mentioning the experiment numbers.
Fig 5: The quality and edition of this figure is worse than in the previous ones.
Figures 3, 4 and 5 are composed by two graphs, they could be numbered or identified with letters. In figures 3 and 5, the graph on the right represents the same information as in figure 6 (accumulated emissions of NH3). Maybe the better figure to represent the accumulated NH3 emissions is figure 6.
Response: Thank you we have now given the diagrams the letters A and B.
Figure 7: The vertical title of the first graph in not visible.
Use the same criterion when placing the statistics letters: close to the error bars, lined up on top or lined at the bottom of the bars.
Response: Thank you the figure has been revised.
Figure 8: Correct the abbreviation of ammonia and nitrate in the vertical title (symbols + and –).
Response: Thank you the symbols + and – (upper case) is included.
Section 3.3: I am missing the results of nitrogen content at the different depths sampled.
Response: We have included the information that 75% of the inorganic N was in found in the layers from 0-30 cm.